# Clinical characteristics of 199 discharged patients with COVID-19 in Fujian Province: A multicenter retrospective study between January 22nd and February 27th, 2020

Sijiao Wang[1,2☯], Zhisheng Chen[1☯], Yijian Lin[3☯], Li Lin[4☯], Qunying Lin[5☯], Sufang Fang[6☯], Yonghong Shi[7], Xibin Zhuang[3], Yuming Ye[4], Ting Wang[6], Hongying Zhang[6]*, Changzhou Shao[1,2]*

1 Department of Pulmonary Medicine, Xiamen Branch, Zhongshan Hospital, Fudan University, Xiamen, Fujian, China, 2 Department of Pulmonary Medicine, Zhongshan Hospital, Fudan University, Shanghai, China, 3 Department of Pulmonary and Critical Care Medicine, Quanzhou First Hospital Affiliated to Fujian Medical University, Quanzhou, Fujian, China, 4 Department of Pulmonary and Critical Care Medicine, Zhangzhou Affiliated Hospital of Fujian Medical University, Zhangzhou, Fujian, China, 5 Department of Pulmonary and Critical Care Medicine, Affiliated Hospital of Putian University, Putian, Fujian, China, 6 Department of Respiratory Medicine, Fuzhou Pulmonary Hospital, Fuzhou, Fujian, China, 7 Department of Respiratory and Critical Care Medicine, the First Affiliated Hospital of Xiamen University, Xiamen, Fujian, China

☯ These authors contributed equally to this work.
* sczzhongshan@126.com (CS); 290923206@qq.com (HZ)

## Abstract

### Background

Coronavirus disease 2019 (COVID-19) has quickly spread throughout the country and the world since first broke out in Wuhan, China. The outbreak that started from January 22, 2020, in Fujian Province has been controlled as the number of indigenous cases has not increased since March. We aimed to describe the clinical characteristics of patients with COVID-19 in Fujian Province, China.

### Methods

In this retrospective, multicenter study, we collected and analyzed the epidemiological, clinical, and laboratory data of all cases confirmed by nucleic acid tests in five designated hospitals in Fujian Province between January 22 and February 27, 2020. All patients were followed up until discharge. COVID-19 severity was classified as mild, moderate, severe, or critical.

### Results

Of 199 discharged patients with COVID-19, 105 patients were male, with a median age of 46.3 years, and 17 patients were severe, and 5 patients were critical on admission. Hypertension and diabetes were the most common comorbidities. The symptoms at illness onset were mainly fever (76.4%), cough (60.8%), and myalgia or fatigue (27.6%). A total of 96.5%

**Data Availability Statement:** All relevant data are within the manuscript.

**Funding:** This work is funded by Fujian Provincial Department of Science and Technology (2020Y01090006). The funders had no role in study design, data collection and analysis, decision to publish, or preparation of the manuscript.

of patients had abnormal imaging findings on chest computed tomography. Lymphopenia (37.2%) and hypoxemia (13.6%) were observed. Acute respiratory distress syndrome and respiratory failure occurred in 9 patients (4.5%) and 8 patients (4.0%) respectively. One patient died and the others were cured and discharged with the median hospital stay of 19 days. Old age was negatively correlated with lymphocyte count (r = - 0.296, p < 0.001) and oxygenation index (r = - 0.263, p = 0.001). Bivariate regression analysis revealed that old age ($\geq$ 75 years), hypertension, diabetes, and lymphopenia were correlated with the severity of COVID-19.

## Conclusions

Patients in Fujian Province were mostly nonsevere cases with mild or moderate symptoms, and had a lower mortality than patients in Wuhan (4.3%-15%). Older age, hypertension, diabetes, and lymphopenia were risk factors for severity of COVID-19.

## Introduction

Coronavirus disease 2019 (COVID-19) has spread throughout China and worldwide, since the first case was confirmed in Wuhan, China in December, 2019 [1–3]. The causative pathogen, severe acute respiratory syndrome coronavirus 2 (SARS-CoV-2), became the third coronavirus after severe acute respiratory syndrome coronavirus (SARS-CoV) and Middle East respiratory syndrome coronavirus (MERS-CoV) that had caused deadly and disruptive epidemic of acute respiratory disease [4, 5]. Next-generation sequencing and phylogenetic analysis suggested that SARS-CoV-2 is a novel beta coronavirus whose original hosts might be bats [6]. SARS-CoV-2 can be transmitted from person to person through respiratory droplets and close contact [7, 8] and has been found in the saliva, feces, and semen of infected patients [9, 10]. As of July 31, 2020, the number of COVID-19 cases had surpassed 17 million globally with 668,910 deaths, and both the Americas and Europe had been the epicenter since the beginning of March [11]. Moreover, more than half of the cases in a skilled nursing facility were asymptomatic or presymptomatic, leading to rapid and widespread transmission which proved that only symptom-oriented strategies were not sufficient to contain the epidemic [12].

Early studies of COVID-19 described that the clinical characteristics were mainly fever, cough, shortness of breath, myalgia or fatigue, and normal or decreased leukocyte counts, and the mortality rate was as high as 15% [13, 14]. Because of old age and underlying comorbidities, older adults were susceptible to severe COVID-19 outcomes such as acute respiratory distress syndrome (ARDS) and death [15–17]. Nevertheless, studies have shown that the symptoms of patients with COVID-19 in Zhejiang Province, Jiangsu Province, Beijing, and Chongqing City were all milder than those of the initially infected patients in Wuhan and other cities of Hubei Province [18–20]. Located in southeastern China, Fujian Province is not adjacent to Hubei Province and has a population of 39.73 million. Since the first patient infected with COVID-19 was confirmed on January 22 in Quanzhou, Fujian, a total of 296 indigenous cases were confirmed in Fujian Province, and the local outbreak was completely controlled due to the number of infected cases kept no increasing since March. At present no study has revealed the comprehensive situation of COVID-19 in Fujian Province.

The purpose of this study was to describe the epidemiological and clinical characteristics of 199 discharged patients diagnosed with COVID-19 during late-January to late-February 2020 in Fujian Province, China. We further explored the potential risk factors associated with

disease severity and hope our findings will provide some details contributing to the understanding of the new highly infectious disease.

## Materials and methods

### Study design and participants

A retrospective, multicenter study enrolled discharged patients who were confirmed and hospitalized in five designated hospitals that responsible for treating patients with COVID-19 in five cities of Fujian Province from January 22 to February 27, 2020, including Fuzhou, Zhangzhou, Xiamen, Putian, and Quanzhou. All confirmed patients were followed-up until they were either discharged or died, thus the follow-up period was exactly the hospital days of the patients. The last patient in this cohort was discharged on March 3.

### Data collection

Nasopharyngeal swab specimens from the upper respiratory tract were obtained from all patients on admission and send to the local Centers for Disease Control to test for SARS-CoV-2 RNA by real time reverse transcription polymerase chain reaction (RT-PCR). Information on demographics (age, sex, body mass index), exposure history (Wuhan exposure history, contact history), clinical characteristics, laboratory results, radiological findings, treatments, and outcomes were collected from electronic medical records. If data were conflicting or missing from records, we contacted directly with patients or their families to ascertain the information. The data were checked by a team of professional physicians.

### Definitions

The patients were clinically classified into four types: mild, moderate, severe, and critical as previously described [20, 21]. Specifically, each type was defined as follows: (a) mild type: patients whose clinical symptoms were mild with no abnormal radiological findings; (b) moderate type: patients who had both pneumonia manifestation on chest computed tomography and clinical symptoms such as fever and cough; (c) severe type: patients who had respiratory rate $\geq$ 30 per min, oxygen saturation without inhaling oxygen at rest $\leq$ 93%, or oxygenation index $\leq$ 300 mmHg (hypoxemia); and (d) critical type: patients who developed respiratory failure requiring mechanical ventilation, shock or organ dysfunction and needed intensive care. If the temperature exceeded 37.3˚C, it was defined as fever. The oxygenation index was defined as arterial oxygen pressure ($PaO_2$) over inspired oxygen fraction ($FiO_2$). The date of illness onset was defined as the day when the first symptom was noticed, and the incubation period was defined as the time from exposure to the illness onset [22]. Familial clusters were identified as the situation in which two or more patients in the same family were infected within two weeks. Once disease progressed into a more serious type, it was recorded as deterioration of the clinical condition. The discharge standards were as follows: temperature kept below 37.3˚C for more than 3 days, respiratory symptoms improved significantly, and nucleic acid test for SARS-CoV-2 was negative for two consecutive times with at least one day interval.

### Statistical analysis

Continuous variables with a normal distribution were presented as the mean (SD) and those with a nonnormal distribution were presented as the median (IQR). Categorical variables were expressed as frequency rates and percentages. All patients were divided into two groups: non-severe (mild and moderate type) cases and severe (severe and critical type) cases, and then bivariate regression analysis of disease severity was performed based on age, sex,

comorbidities, and laboratory parameters. SPSS (version 22.0) was used for the above analyses. Spearman's correlation test was used for analyzing the correlations between age and lymphocyte count as well as oxygenation index with GraphPad Prism version 8.0 software. A *P* value < 0.05 was considered statistically significant.

### Ethics statement

This study was approved by the ethical committee in Zhongshan Hospital, Xiamen Branch, Fudan University (B2020-003). The requirement for informed consent was waived because the data were urgently collected and analyzed anonymously.

## Results

### Epidemiological and clinical characteristics

There were 199 patients infected with SARS-CoV-2 in five hospitals from January 22 to February 27. Of the 199 patients, the median age was 46.3 (SD 16.4, range 16–93; Table 1) years and 105 patients (52.8%) were men. A total of 114 patients had a residing or short traveling history in Wuhan, and one of them had direct exposure to the Huanan seafood market. Twenty-eight patients had close contact with people from Wuhan. Sixty-eight patients were involved in family clusters, with the largest one including six COVID-19 cases. Over one-third of patients had underlying diseases including hypertension (15.6%), diabetes (7.5%), cardiovascular disease (5.5%), and respiratory disease (5.5%) such as tuberculosis, pulmonary bulla, asthma, and chronic obstructive lung disease. Eleven patients (5.5%) complicated with chronic liver disease had a history of hepatitis B infection. The median incubation period was 6 (IQR 3–10) days among 115 patients offering the exact contact date.

Except for common symptoms like fever, cough and myalgia/fatigue, gastrointestinal symptom such as diarrhea occurred in 17 patients (8.5%), and the typical upper respiratory tract symptoms such as rhinorrhea and nasal congestion were present in 15 patients (7.5%). Moreover, one patient developed hyposmia at illness onset, and eight presymptomatic patients were admitted to the hospital due to the contact with confirmed cases and had symptoms during hospitalization. The severity classification of different age groups was shown in Fig 1A. Older patients occupied a higher proportion of severe/critical type than younger patients. Furthermore, all patients had a chest CT examination, 81.4% of whom showed bilateral abnormalities and 67.8% of whom presented ground-glass opacities as shown in Fig 2.

### Laboratory and radiological characteristics

Routine blood tests on admission showed that white blood cell count lower than the normal range was present in 40 patients (20.1%) and lymphopenia was found in 74 patients (37.2%) (Table 2). Seventeen patients (8.5%) had decreased platelets (less than $125 \times 10^9$/L) and 14 patients (7.0%) had hemoglobin levels lower than normal range (150 g/L). There were 27 patients (13.6%) with high levels of procalcitonin (more than 0.1 ng/mL) and 82 patients (41.2%) with high level of C- reactive protein (over 10 mg/L).

Regarding hepatic and renal function, 26 patients (13.1%) showed lower level of albumin than normal range (Table 2). The alanine transaminase (ALT) level of 22 patients (11.1%) and aspartate aminotransferase (AST) levels of 47 patients (23.6%) were above the normal range. Sixty-five patients (32.7%) had increased levels of lactate dehydrogenase (LDH). Only 14 patients (7.0%) and 16 patients (8.0%) had increased blood urea nitrogen and serum creatinine respectively, and 22 patients (11.1%) had increased creatine kinase. There were 10 patients with the prolonged prothrombin time and 29 patients with increased D-dimer levels. In terms

**Table 1. Demographic, epidemiological and clinical features of patients with COVID-19 in Fujian Province.**

| Variables | Patients (n = 199) |
|---|---|
| **Age, years** | |
| Mean (SD) | 46.3 (16.4) |
| Range | 16–93 |
| 16–24 | 12 (6.0) |
| 25–39 | 72 (36.2) |
| 40–54 | 59 (29.6) |
| 55–69 | 35 (19.6) |
| ≥70 | 21 (10.6) |
| **BMI** | 23.8 (21.2–26.0) (n = 173) |
| Low weight (<18.5) | 7 (3.5) |
| Normal weight (18.5–24.9) | 106 (53.3) |
| Overweight (25–29.9) | 53 (26.6) |
| Obese (>30) | 7 (3.5) |
| **Sex** | |
| Male | 105 (52.8) |
| **Current smoking** | 13 (6.5) |
| **History of exposure** | |
| Having been to Wuhan in last 2 weeks | 114 (57.3) |
| Contact with people from Wuhan | 28 (14.1) |
| No relation with Wuhan | 53 (26.6) |
| Family clusters | 68 (34.2) |
| **Days of incubation period** | 6 (3–10) (n = 115) |
| **Any Comorbidities** | 73 (36.7) |
| Hypertension | 31 (15.6) |
| Diabetes | 15 (7.5) |
| Respiratory disease | 11 (5.5) |
| Chronic liver disease | 11 (5.5) |
| Cardiovascular disease | 8 (4.0) |
| Malignant tumor | 8 (4.0) |
| Chronic kidney disease | 4 (2.0) |
| Nervous system diseases | 3 (1.5) |
| **Signs and symptoms** | |
| Heart rate, bpm | 86 (78–94) |
| Mean arterial pressure | 96 (89–104) |
| Fever | 152 (76.4) |
| Highest temperature, ˚C | |
| <37.3 | 13 (6.5) |
| 37.3–38.0 | 77 (38.7) |
| 38.1–39.0 | 61 (30.7) |
| >39.0 | 7 (3.5) |
| Cough | 121.0 (60.8) |
| Myalgia or fatigue | 55 (27.6) |
| Chest distress | 18 (9.0) |
| Chills | 18 (9.0) |
| Panting | 17 (8.5) |
| Diarrhea | 17 (8.5) |
| Rhinorrhea | 15 (7.5) |

(*Continued*)

**Table 1.** (Continued)

| Variables | Patients (n = 199) |
|---|---|
| Dizziness | 13 (6.5) |
| Sore throat | 13 (6.5) |
| Dyspnoea | 2 (1.0) |
| Hyposmia | 1 (0.5) |
| **Clinical classifications** | |
| Mild type | 7 (3.5) |
| Moderate type | 170 (85.4) |
| Severe type | 17 (8.5) |
| Critical type | 5 (2.5) |
| **Chest CT findings at illness onset** | |
| Pneumonia | 192 (96.5) |
| Bilateral distribution | 162 (81.4) |
| Ground-glass opacity | 135 (67.8) |

The data are mean (SD) or n (%). COVID-19 = coronavirus disease 2019, CT = computed tomography.

of blood gas analysis, 27 patients with decreased arterial oxygen saturation ($SaO_2$) and oxygenation index had significant hypoxemia. In addition, older age was correlated with the lower lymphocytes count (r = -0.296, p < 0.001, Fig 1B) and was also associated with lower oxygenation index (r = -0.263, p = 0.001, Fig 1C).

One patient did not display positive nucleic acid results for SARS-CoV-2 until the fifth detection. In addition to the test for SARS-CoV-2, all patients were tested for the nucleic acid, IgM or IgA of nine kinds of respiratory pathogens, including influenza A and B. Culture of bacteria and fungi were conducted in patients with fever. The results showed that two patients were positive for the nucleic acid of influenza B, three were positive for influenza B IgM, and another seven patients were positive for mycoplasma IgM (Table 2).

## Treatments and prognosis

During hospitalization, patients had received antiviral treatment, interferon inhalation, empirical antibiotics, and Chinese medicine, accounting for 100%, 52.8%, 48.2%, and 40.2%,

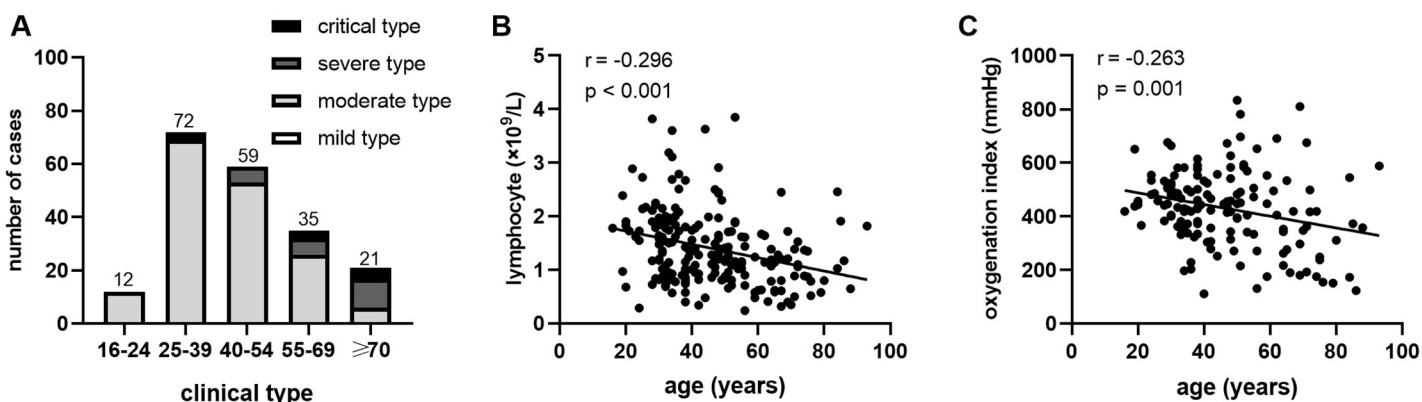

**Fig 1. Age distributions, clinical classifications and correlations of age, lymphocyte count, and oxygenation index in patients with COVID-19.** In different age groups, (A) number of confirmed patients and the distribution of four clinical classifications, and (B) correlations between age and blood lymphocyte number and (C) oxygenation index on admission in all patients.

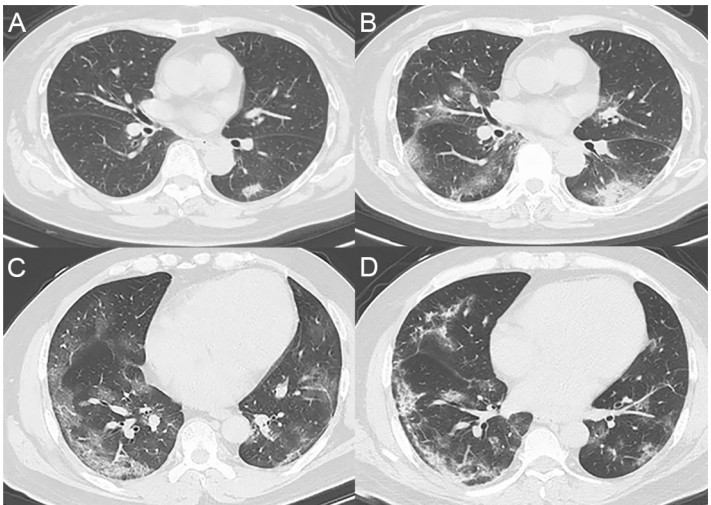

**Fig 2. Representative thoracic CT images.** (A) CT images from a 71-year-old woman showing bilateral focal ground-glass opacities (GGOs) close to subpleural at day 3 after illness onset and (B) multiple patchy shadows with increasing GGOs at day 10 after illness onset. (C) CT images of a 48-year-old man showing bilateral patchy shadows and multiple GGOs at day 9 after illness onset; (D) the GGOs were absorbed mostly leaving linear opacities or fibrous stripes at day 18 after illness onset.

respectively (Table 3). Thirty- four patients were administered corticosteroids and 29 patients received intravenous immunoglobulin. Oxygen therapy was given to 83 patients, and only one patient who had ever received a left kidney transplantation underwent extracorporeal membrane oxygenation (ECMO).

The patients were divided into two groups: the non-severe group including mild and moderate type cases, and the severe group including severe and critical type cases (Table 3). Sixteen patients developed deterioration during hospitalization, and the final classification was 166 moderate type, 24 severe type and 9 critical type patients. Of 33 patients in the severe group, 15 patients developed complications including ARDS, respiratory failure, multiple systemic organ failure, and shock, and 14 patients were admitted to intensive care unit (ICU). Moreover, two thirds of severe group had evidence of bacterial infections. By the end of the follow-up, only an 84-year-old man complicated with hypertension and cerebral infraction died on the ninth day of hospitalization, and the remaining patients were cured and discharged. The median period from admission to discharge was 19 (IQR 14–25) days and the duration from the first positive to the last negative RT-PCR for SARS-CoV-2 was 15 (IQR 11–21) days.

Additionally, binary logistic regression analysis was conducted in two groups (Table 4), and found that age $\geq$ 75 years (odds ratio, 5.61; 95% CI, 1.16–27.11; p = 0.032), hypertension (3.44; 1.06–11.16; p = 0.040), diabetes (6.94; 1.65–29.22; p = 0.008), and lymphocyte count (0.27; 0.091–0.782; p = 0.016) were associated with a high risk of developing severe COVID-19, while total bilirubin (0.993; 0.919–1.073; p = 0.863) and serum creatinine (1.019; 0.982–1.056; p = 0.323) were not risk factors of COVID-19.

## Discussion

In this multicenter case series study, we reported 199 patients who were diagnosed with COVID-19 and treated in a timely manner in Fujian Province, depicting the status of early SARS-CoV-2 outbreaks in five cities of Fujian Province. The proportion of critical patients was 17% - 32%, and the mortality ranged from 4.3% to 15% in early reports from Wuhan [13–

**Table 2. Laboratory findings of patients with COVID-19 on admission in Fujian Province.**

| | Normal range | All patients (n = 199) | | |
|---|---|---|---|---|
| | | Median (IQR) | Increased No (%) | Decreased No (%) |
| **Blood routine** | | | | |
| White blood cell count, ×10$^9$/L | 3.5–9.5 | 5.03 (3.79–6.45) | 10 (5.0) | 40 (20.1) |
| Neutrophil count, × 10$^9$/L | 1.8–6.3 | 3.24 (2.21–4.39) | 18 (9.0) | - |
| Lymphocyte count, × 10$^9$/L | 1.1–3.2 | 1.28 (0.89–1.78) | - | 74 (37.2) |
| Platelet count, × 10$^9$/L | 125.0–350.0 | 191 (156–238) | 9 (4.5) | 17 (8.5) |
| Hemoglobin, g/L | 115–150 | 137 (126–149) | - | 14 (7.0) |
| **Blood biochemistry** | | | | |
| Albumin g/L | 35.0–50.0 | .40.3 (36.8–44.3) | - | 26 (13.1) |
| Total bilirubin, μmol/L | 3.0–22.0 | 12.9 (8.2–18.4) | 34 (17.1) | - |
| Alanine aminotransferase, U/L | 9.0–52.0 | 24 (17–34) | 22 (11.1) | - |
| Aspartate aminotransferase, U/L | 14.0–36.0 | 25 (20–32) | 47 (23.6) | - |
| Lactate dehydrogenase, U/L | 140–350 | 225 (169–407) | 65 (32.7) | - |
| Blood urea nitrogen, mmol/L | 2.5–6.1 | 3.7 (3.0–4.6) | 14 (7.0) | - |
| Serum creatinine, μmol/L | 53.0–97.0 | 67 (55–79) | 16 (8.0) | - |
| Creatine kinase, U/L | 38.0–174.0 | 65 (43–107) | 22 (11.1) | - |
| Potassium, mmol/L | 3.5–5.1 | 3.9 (3.6–4.2) | 3 (1.5) | 33 (16.6) |
| Sodium, mmol/L | 137.0–147.0 | 139 (136–140) | 2 (1.0) | 52 (26.1) |
| **Coagulation function (n = 170)** | | | | |
| Prothrombin time | 10.5–13.5 | 11.8 (11.2–12.5) | 10 (5.0) | 13 (6.5) |
| Activated partial thromboplastin time | 22.0–38.0 | 29.9 (27.3–33.7) | 17 (8.5) | 3 (1.5) |
| Fibrinogen | 2.0–4.0 | 3.5 (3.0–4.3) | 53 (26.6) | 3 (1.5) |
| D-dimer | 0–0.5 | 0.23 (0.05–0.38) | 29 (14.6) | - |
| **Blood gas analysis (n = 153)** | | | | |
| pH | 7.35–7.45 | 7.42 (7.39–7.44) | 24 (12.1) | 4 (2.0) |
| PaO$_2$, mmHg | 83.0–108.0 | 95.6 (80.5–111.0) | - | 40 (20.1) |
| SaO$_2$, mmHg | 95.5–98 | 97.7 (96.1–98.2) | - | 27 (13.6) |
| PaO$_2$:FiO$_2$, mmHg | >300 | 438 (342–512) | - | 27 (13.6) |
| **Infection-related biomarkers** | | | | |
| Procalcitonin, ng/mL | 0–0.1 | 0.04 (0.03–0.06) | 27/187 | - |
| C-reactive protein, mg/L | <10 | 6.74 (4.68–21.6) | 82/181 | - |
| Nucleic acid positive for influenza B | - | 2 (0.02) | - | - |
| IgM positive for influenza B | - | 3 (0.02) | - | - |
| IgM positive for mycoplasma | - | 7 (0.04) | - | - |

The data are the median (IQR), n% or n/N. IQR = interquartile range, PaO$_2$ = arterial oxygen pressure, SaO$_2$ = arterial oxygen saturation, FiO$_2$ = inspired oxygen fraction.

15, 17]. In our study, only 33 patients (16.6%) exhibited severe illness and one died, suggesting that the early outbreak in Fujian Province may have been milder than that in Wuhan. Moreover, as of February 27, based on the data published by the CDC, 296 indigenous cases of SARS-CoV-2 infection and one death were reported in Fujian Province, which were far fewer than the 48 137 cases and 2132 deaths reported in Wuhan. Consistent with the findings of previous studies, our results also showed that hypertension and diabetes were risk factors for COVID-19 [15, 23].

One third of the infected patients were related to family clusters in this study, suggesting that avoiding intrafamilial transmission is urgent to control the pandemic. In addition, the

**Table 3. Treatments and outcomes of patients with COVID-19 during hospitalization.**

| | All patients (n = 199) | Non-severe (n = 166) | Severe (n = 33) |
|---|---|---|---|
| **Treatments** | | | |
| Antiviral therapy | 199 (100) | 166 (100) | 33 (100) |
| Lopinavir/ritonavir | 192 (96.5) | 163 (98.2) | 31 (93.9) |
| Arbidol | 72 (36.2) | 54 (32.5) | 18 (54.4) |
| Ribavirin | 36 (18.1) | 33 (19.9) | 3 (9.1) |
| Oseltamivir | 11 (5.5) | 7 (4.22) | 4 (12.1) |
| Interferon inhalation | 105 (52.8) | 87 (52.4) | 18 (54.5) |
| Antibiotics therapy | 96 (48.2) | 68 (41) | 28 (84.8) |
| Chinese medicine | 80 (40.2) | 66 (39.8) | 14 (42.4) |
| Corticosteroid | 34 (17.1) | 17 (10.2) | 17 (51.5) |
| Intravenous immunoglobulin | 29 (14.6) | 13 (7.8) | 16 (48.5) |
| Oxygen therapy | 83 (41.7) | 56 (33.7) | 27 (81.8) |
| Nasal cannula | 70 (35.2) | 55 (33.1) | 15 (45.5) |
| Non-invasive mechanical ventilation (high-flow nasal cannula or face mask) | 8 (4.0) | 1 (0.6) | 7 (21.2) |
| Invasive mechanical ventilation | 4 (2.0) | - | 4 (12.1) |
| ECMO | 1 (0.5) | - | 1 (3.0) |
| **Complications** | 15 (7.5) | 0 | 15 (45.5) |
| ARDS | 9 (4.5) | 0 | 9 (27.3) |
| Respiratory failure | 8 (4.0) | 0 | 8 (24.2) |
| Multiple systemic organ failure | 3 (1.5) | 0 | 3 (9.1) |
| Shock | 2 (1.0) | 0 | 2 (6.1) |
| Admission to intensive care unit | 14 (7.0) | 0 | 14 (42.4) |
| Complicated with bacterial infection | 38 (19.1) | 16 (9.6) | 22 (66.7) |
| Deterioration | 16 (8.0) | 0 | 16 (48.5) |
| **Prognosis** | | | |
| Survival and discharge | 198 (99.5) | 166 | 32 (97) |
| Death | 1 (0.5) | 0 | 1 (3.0) |
| Days from first positive to last negative RT-PCR | 15 (11–21) | 15 (11–20) | 18 (14–23) |
| Days from admission to discharge | 19 (14–25) | 19 (15–24) | 24 (20.5–28) |

The Data are n (%) or median (IQR). ECMO = extracorporeal membrane oxygenation.

transmission of SARS-CoV-2 through asymptomatic carriers via person-to-person contact had been observed in many reports [7, 8, 12, 24]. Asymptomatic carriers were often diagnosed by screening after other family members or close contacts developed symptoms, and they were

**Table 4. Binary logistic regression analysis of risk factors of severe COVID-19.**

| Characteristics and findings | OR (95%CI) | p value |
|---|---|---|
| Age ($\geq$ 75 years vs. < 75) | 5.608 (1.160–27.109) | 0.032 |
| Sex (female vs. male) | 0.449 (0.114–1.774) | 0.254 |
| Hypertension (Yes vs. No) | 3.436 (1.058–11.162) | 0.040 |
| Diabetes (Yes vs. No) | 6.936 (1.646–29.220) | 0.008 |
| Cardiovascular disease (Yes vs. No) | 2.509 (0.454–13.851) | 0.291 |
| Lymphocyte count ($\times 10^9$ /L) | 0.267 (0.091–0.782) | 0.016 |
| Serum creatinine (U/L) | 1.019 (0.982–1.056) | 0323 |
| Total bilirubin (μmol/L) | 0.993 (0.919–1.073) | 0.863 |

an important cause of the COVID-19 pandemic [25, 26]. Moreover, presymptomatic transmission and asymptomatic infection displayed a strong ability to spread the virus [12, 24], exerting great difficulty in restraining the pandemic, so strategies focused solely on clinical symptoms may be not sufficient to prevent the transmission and avoid resurgence of SARS-CoV-2.

Early studies of COVID-19 in Wuhan showed that female patients accounted for 32% - 45.7% [14, 15, 21, 27], but no obvious difference in gender proportion was found in our study. Old age was associated with higher risk of developing ARDS and a lower survival rate than the young, and hypertension was associated with a high risk of mortality from COVID-19 [17, 21, 23]. Consistently, our bivariate logistic regression analysis also revealed that age over 75 years and underlying disease (hypertension and diabetes) were the risk factors for COVID-19 severity. Furthermore, we found that old age was negatively associated with lymphocyte count and oxygenation index, implying that old age may be a dependent risk factor for progression.

Pathological findings had revealed that counts of peripheral $CD4^+$ T and $CD8^+$ T cells were substantially decreased in COVID-19 patients with ARDS [28], suggesting that SARS-CoV-2 may exert major impact on lymphocytes, especially T lymphocytes. Another study showed that critical patients had the lowest percentage of $CD8^+$ T cells among four types of patients with COVID-19 [29]. In addition, laboratory parameters such as lymphocytes could predict the progression of COVID-19 [30]. Here, the binary logistic regression model revealed that low lymphocyte count was an important risk factor for progressing into a severe/critical type of COVID-19.

In this cohort, 5.5% of cases were complicated with chronic hepatitis B, and both ALT and AST levels were increased in some patients. It was suggested that liver damage was often transient in mild cases but prevalent in severe cases of COVID-19 [31]. Moreover, SARS-CoV-2 infected the host via the angiotensin-converting enzyme 2 (ACE2) receptor, which was enriched in cholangiocytes except for lungs, heart, kidney, and intestines [20]. However, the mechanisms of liver damage caused by SARS-CoV-2 require further study.

Approximately half of the severe patients developed complications such as ARDS, respiratory failure, and shock, and were always administered corticosteroids and intravenous immunoglobulins. According to the WHO recommendations published on May 27th, 2020, systemic corticosteroid treatment was not routinely recommended for COVID-19 patients unless they had indications of exacerbation of asthma or chronic obstructive pulmonary disease, septic shock or ARDS [32]. However, studies of corticosteroids for the novel coronavirus pneumonia have yielded various findings. One case-control study showed that methylprednisolone treatment could effectively improve symptoms but prolong the negative conversion of nucleic acids [33]. The Randomized Evaluation of COVID-19 Therapy (RECOVERY) trial found that dexamethasone reduced the 28-day all-cause mortality and was especially beneficial to patients who had symptoms for more than 7 days or required mechanical ventilation [34]. Moreover, a prospective meta-analysis of seven randomized clinical trials of critically ill patients with COVID-19 also showed that the administration of corticosteroids was associated with lower 28-day all-cause mortality [35]. However, considering that high dose corticosteroid use was closely associated with adverse events [36], the benefits and risks must be carefully weighed before commencing corticosteroid therapy, and the dosage and duration should be evaluated prudently.

All patients received the antiviral drug lopinavir/ritonavir in the cohort, nevertheless, the first trial of lopinavir/ritonavir in adults with severe COVID-19 showed no benefit in contrast with patients receiving standard care [37]. Remdesivir, a promising antiviral drug for SARS-CoV-2, was reported that could shorten the recovery time from 15 days to 11 days in a double-blind, randomized, placebo-controlled trial involving 1063 adults [38]. Moreover, more than

150 candidates are under development since human vaccines for SARS-CoV-2 are not currently available.

The study naturally has several limitations. First, since the data were collected from different hospitals, the reference values of partial laboratory parameters varied greatly. Second, a total of 296 indigenous patients were confirmed and hospitalized in nineteen designated hospitals for SARS-CoV-2 in Fujian Province during the epidemic. Herein, 199 patients with COVID-19 from five hospitals were enrolled, so our conclusions may only represent the epidemic situation in Fujian Province to some extent. Last, the relatively short period of follow-up was limited to the hospital stay, which prevented us from further assessing the readmission, deterioration and sequela in those discharged patients as continuous follow-up of discharged patients is necessary and indispensable to study the long-term effects of COVID-19 on human health.

In summary, our study firstly suggested that compared with the COVID-19 patients in Wuhan during the early outbreak, most COVID-19 patients in Fujian Province were nonsevere cases with a relatively lower mortality rate. In addition, old age, comorbidities such as hypertension and diabetes, and lymphopenia were potential risk factors for patients with COVID-19 to progress into a severe/critical type.

## Acknowledgments

We greatly appreciate the kind assistance of Prof. Yuanlin Song from Zhongshan Hospital, Fudan University for guidance in study design.

## Author Contributions

**Conceptualization:** Sijiao Wang, Zhisheng Chen, Yijian Lin, Li Lin, Qunying Lin, Hongying Zhang, Changzhou Shao.

**Data curation:** Sijiao Wang, Zhisheng Chen, Yijian Lin, Li Lin, Qunying Lin, Sufang Fang, Yonghong Shi, Xibin Zhuang, Yuming Ye, Ting Wang.

**Formal analysis:** Sijiao Wang, Yijian Lin, Li Lin, Sufang Fang, Xibin Zhuang, Yuming Ye, Hongying Zhang.

**Funding acquisition:** Changzhou Shao.

**Investigation:** Sijiao Wang, Zhisheng Chen, Yijian Lin, Li Lin, Qunying Lin, Sufang Fang, Yonghong Shi, Yuming Ye, Ting Wang, Hongying Zhang.

**Methodology:** Sijiao Wang, Zhisheng Chen, Yijian Lin, Yonghong Shi, Xibin Zhuang, Yuming Ye, Ting Wang, Changzhou Shao.

**Project administration:** Yijian Lin, Hongying Zhang, Changzhou Shao.

**Resources:** Li Lin, Qunying Lin, Sufang Fang, Yonghong Shi, Yuming Ye, Ting Wang, Hongying Zhang, Changzhou Shao.

**Software:** Sijiao Wang, Zhisheng Chen, Xibin Zhuang.

**Supervision:** Sijiao Wang, Zhisheng Chen, Li Lin, Qunying Lin, Sufang Fang, Hongying Zhang, Changzhou Shao.

**Validation:** Sijiao Wang, Zhisheng Chen, Yijian Lin, Li Lin, Qunying Lin, Sufang Fang, Yonghong Shi, Yuming Ye, Hongying Zhang, Changzhou Shao.

**Visualization:** Sijiao Wang, Zhisheng Chen, Yijian Lin, Li Lin, Qunying Lin, Sufang Fang, Yonghong Shi, Xibin Zhuang, Ting Wang, Hongying Zhang, Changzhou Shao.

**Writing – original draft:** Sijiao Wang, Zhisheng Chen.

**Writing – review & editing:** Sijiao Wang, Zhisheng Chen, Yijian Lin, Li Lin, Qunying Lin, Sufang Fang, Yonghong Shi, Xibin Zhuang, Yuming Ye, Hongying Zhang, Changzhou Shao.

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
