## [Decision Letter · Decision Letter 0]

14 Jul 2020

PONE-D-20-18174

Clinical characteristics of 199 discharged patients with COVID-19 in Fujian province: a multicenter retrospective study

PLOS ONE

Dear Dr. shao,

Thank you for submitting your manuscript to PLOS ONE. After careful consideration, we feel that it has merit but does not fully meet PLOS ONE’s publication criteria as it currently stands. Therefore, we invite you to submit a revised version of the manuscript that addresses the points raised during the review process.

The manuscript describes important findings from hospitalized cases of COVID-19 in Fujian province at the beginning of the COVID-19 outbreak there. The manuscript could benefit from review by review for grammar and flow in English. In general, in additional to the minor comments below, the discussion section should be modified so that the conclusions more accurately reflect the results and the relativelhy small sample size.

Please completely address the comments below and the comments by the reviewer before submitting the revised manuscript, and please ensure that the update manuscript has been reviewed for grammar and flow in English before submission. 

Abstract:<o:p></o:p>

Intro: The authors should add some background information on the COVID-19 outbreak in China and when the outbreak started in Fujian, and what is known until now about the outbreak in Fujian.<o:p></o:p>

Methods: The authors should describe what was done in the methods but the number of patients should be first introduced in the results section. In the methods section please include the dates of the study and explain during what part of the outbreak the data were collected. Were these the first hospitalized cases in the province or was this a convenience sample from the middle of the outbreak?<o:p></o:p>

Results: Please define severe and critical in the methods section. Line 35 please describe more clearly the relationship between age and lymphopenia and oxygen. The older the person the lower the lymphocyte count and the lower the oxygen level? Please describe this in the text.<o:p></o:p>

Conclusions: Mention the findings about severity in Wuhan so that the reader can understand what was different here.<o:p></o:p>

<o:p> </o:p>

Manuscript:<o:p></o:p>

<o:p> </o:p>

Title: Add dates of the study<o:p></o:p>

<o:p> </o:p>

Intro:<o:p></o:p>

Please briefly the history of the COVID-19 epidemic of Fujian province and also the characteristics of the provine. When was the first case identified. How many cases have been identified until now? Where is Fujian province in China in relation to Wujan and what is the population?<o:p></o:p>

<o:p> </o:p>

Methods:<o:p></o:p>

<o:p> </o:p>

Line 78. The ethics statement should be placed at the end of the methods section.<o:p></o:p>

<o:p> </o:p>

Definitions.<o:p></o:p>

Please provide the definition for moderate patients.<o:p></o:p>

<o:p> </o:p>

Please explain how fever was defined? (What temperature cutoff?)<o:p></o:p>

<o:p> </o:p>

Please explain what the follow-up period was. Did everyone get follow-up until they were either discharged from the hospital or died?<o:p></o:p>

<o:p> </o:p>

Results.<o:p></o:p>

Please explain whether the 199 patients included all of the patients who were hospitalized at these five hospitals or whether this was a convenience sample.<o:p></o:p>

<o:p> </o:p>

Line 124. Please provide a breakdown of BMI by normal weight, overweight, and obese<o:p></o:p>

<o:p> </o:p>

Table 1.<o:p></o:p>

Mild/common/severe/critical: Do the authors mean moderate rather than common?<o:p></o:p>

<o:p> </o:p>

Did all 199 patients get a CT? Please add this information.<o:p></o:p>

<o:p> </o:p>

Line 138 and line 144. Please add a percentage to all numbers.<o:p></o:p>

<o:p> </o:p>

Line 151. Please phrase this association more clearly – was older age associated with lower lymphocyte counts?<o:p></o:p>

<o:p> </o:p>

Line 161. Chinese medicine like Lianhuaqingwen accounted for 100%, 52.8%, 48.2%, and 40.2%, respectively. This sentence is not clear. Please rephrase.<o:p></o:p>

<o:p> </o:p>

Line 167. Aggravation – this term should be “deterioration in their clinical condition.” Aggravation does not seem like the appropriate word in English.<o:p></o:p>

<o:p> </o:p>

Did any patients die? Please add this information to the results section.<o:p></o:p>

<o:p> </o:p>

Table 4. Please describe the comparison for lymphocyte count, bilirubin and creatinine.<o:p></o:p>

<o:p> </o:p>

Discussion<o:p></o:p>

<o:p> </o:p>

Line 183<o:p></o:p>

Please provide a specific comparison with numbers and percentages to support the claim that there were less severe patients and less deaths in the Fujian cohort compared to Wuhan.<o:p></o:p>

<o:p> </o:p>

Also, I am not convinced that a cohort of 199 patients is large enough to draw comparisons to the Wuhan about disease severity.<o:p></o:p>

<o:p> </o:p>

In addition, the authors’ conclusions about early detection and quarantine measures being responsible for less severity are not supported by this article. The authors do not discuss incidence in this paper, and the authors also do not at all discuss quarantine measures that were implemented in Fujian province and how those compare to Wuhan. Finally, the cohort of 200 patients is quite small.<o:p></o:p>

<o:p> </o:p>

I suggest removing these broad conclusions and remaining with more modest conclusions.<o:p></o:p>

<o:p> </o:p>

Line 192.<o:p></o:p>

What is the connection of intestinal spread to family clusters? This is not logical. I suggest refocusing this paragraph on transmission and perhaps asymptomatic and presymptomatic transmission. The authors could cite other articles that described that most transmission occur in families.<o:p></o:p>

<o:p> </o:p>

Line 197.<o:p></o:p>

The English needs to be reworked here<o:p></o:p>

<o:p> </o:p>

Line 214.<o:p></o:p>

The authors should present data about Hepatitis B prevalence among cases in the results section if they intend to reference these data in the discussion section.<o:p></o:p>

<o:p> </o:p>

Line 220.<o:p></o:p>

Please update this paragraph to reflect the more recent findings that dexamethasone reduced mortality in severe COVID-19 patients and the changes in recommendations.<o:p></o:p>

<o:p> </o:p>

Line 228.<o:p></o:p>

There has been evidence of effectiveness of an antiviral drug, remdesivir, in reducing time to hospital discharge. The authors should update this paragraph to reflect these findings.<o:p></o:p>

<o:p> </o:p>

Line 234.<o:p></o:p>

The authors should not introduce new information in the discussion section. Information about course of virus positivity should be introduced in the results section and commented on in the discussion section.<o:p></o:p>

<o:p> </o:p>

Limitations<o:p></o:p>

The authors should speak about how representative these hospitals are of hospitals in Fujian province. How many hospitals are there in Fujian? How many hospitalized cases of COVID-19 were there during the same period in other hospitals in Fujian?<o:p></o:p>

<o:p> </o:p>

Conclusions paragraph<o:p></o:p>

The authors should be cautious about drawing conclusions relative to Wuhan because of the small sample size.<o:p></o:p>

<o:p> </o:p>

We look forward to receiving your revised manuscript.

Kind regards,

Mark Katz

Academic Editor

PLOS ONE

Journal Requirements:

"This work is funded by Fujian Provincial Department of Science and Technology (2020Y01090006)."

3. We note you have included a table to which you do not refer in the text of your manuscript. Please ensure that you refer to Table 3 in your text; if accepted, production will need this reference to link the reader to the Table.

Reviewers' comments:

Reviewer's Responses to Questions

**Comments to the Author**

1. Is the manuscript technically sound, and do the data support the conclusions?

Reviewer #1: Yes

2. Has the statistical analysis been performed appropriately and rigorously? 

Reviewer #1: Yes

3. Have the authors made all data underlying the findings in their manuscript fully available?

Reviewer #1: Yes

4. Is the manuscript presented in an intelligible fashion and written in standard English?

Reviewer #1: Yes

5. Review Comments to the Author

Reviewer #1: This manuscript is a descriptive retrospective analysis of 199 COVID-19 patients admitted to one of five tertiary care centers in Fujian province, China between January 22 and February 27, 2020. A secondary aim of the study was to identify factors associated with disease severity. Study inclusion criteria and statistical analyses are appropriate. The data collection represents a relatively short period of observation, which may influence some of the study findings. Authors state that all data are available in the manuscript and additional files.

Methods:

Additional description of how severity categories were collapsed for multivariable logistic regression is warranted in the methods section. It appears that severe/critical cases were combined and compared to mild/moderate severity illnesses. But this should be explicitly stated.

Results:

In the description of treatment (text and table) it might be of interest to report which antivirals were used for treatment I see this is mentioned in the discussion, but could be noted in the results as well.

Discussion:

On page 17 line 244-245 the authors state "compared with patients infected with COVID-19 in Wuhan, our study suggested most patients in Fujian province were mild and moderate with a minority of severe cases". I think this overstates the results. In this relatively small study of hospitalized patients for a short period (4 weeks) there were 33/199 (16.6%) that were severe, which is in line with estimates from other locations. Case fatality was lower for the observation period in this study compared to others. I'm not convinced the follow-up was long enough to accurately determine the true extent of severe illness and mortality (as data collection ended on March 3, 2020). Readmission and deterioration have been issues in other places. Other the conclusions follow from the results. Some discussion of the implications of the short follow-up time is warranted.

6. PLOS authors have the option to publish the peer review history of their article (what does this mean?). If published, this will include your full peer review and any attached files.

Reviewer #1: No

---

## [Author Response · Author response to Decision Letter 0]

3 Sep 2020

Dear Editors and Reviewers,

Thank you for your letter and for the reviewers’ comments concerning our manuscript entitled “Clinical characteristics of 199 discharged patients with COVID-19 in Fujian province: a multicenter retrospective study” (ID: PONE-D-20-18174). Those comments are all valuable and very helpful for revising and improving our paper, as well as the important guiding significance to our researches. We have studied comments carefully and have made correction which we hope would meet with approval. Revised portion are marked in yellow in the paper. The main corrections in the paper and responds to reviewer’s comments are as following:

Abstract:

Intro: The authors should add some background information on the COVID-19 outbreak in China and when the outbreak started in Fujian, and what is known until now about the outbreak in Fujian.

Methods: The authors should describe what was done in the methods but the number of patients should be first introduced in the results section. In the methods section please include the dates of the study and explain during what part of the outbreak the data were collected. Were these the first hospitalized cases in the province or was this a convenience sample from the middle of the outbreak?

Results: Please define severe and critical in the methods section. Line 35 please describe more clearly the relationship between age and lymphopenia and oxygen. The older the person the lower the lymphocyte count and the lower the oxygen level? Please describe this in the text.

Conclusions: Mention the findings about severity in Wuhan so that the reader can understand what was different here.

 Reply: Thank you for your kind comments. We have added the background information of COVID-19 outbreak in China and Fujian province (Page 2, line 24-26). The dates of the study was added and the number of patients was deleted in Methods section (Page 2, line 30-31), and all cases were the first hospitalized cases in these cities (Page 2, line 29). The definition of four types of COVID-19 was introduced in Methods section in manuscript (Page 6, line 96-102), and the description of relationship between age and lymphopenia and oxygen was rephrased clearly (Page 2, line 39-40). The findings about mortality in Wuhan was mentioned in the Conclusions (Page 3, line 43).

Manuscript:

Title: Add dates of the study

 Reply: Thanks for your serious comments. We have added dates of the study into the title. (Page 1, line 2-3)

Intro:

Please briefly the history of the COVID-19 epidemic of Fujian province and also the characteristics of the provine. When was the first case identified. How many cases have been identified until now? Where is Fujian province in China in relation to Wuhan and what is the population?

 Reply: Thanks for your comments. According to data published by the Centers for Disease Control of Fujian province, we introduced briefly when and where the first COVID-19 case was confirmed in Fujian, and the total number of indigenous cases (Page 4, line 67-69). We also described the geographical location and population of Fujian province (Page 5, line 69-71)

Methods:

Line 78. The ethics statement should be placed at the end of the methods section.

 Reply: Thanks for your suggestion. We have placed the ethics statement at the end of methods section behind the Statistical Analysis. (Page 7, line 118-121)

Definitions.

Please provide the definition for moderate patients.

 Reply: Thanks for your reminding. We apologize for our negligence regarding confusing the moderate patients with common type. We have corrected “common type” as “moderate type” and reintroduce the definition of four types of COVID-19 patients respectively. In addition, we also revised that in Fig 1A (Page 6, line 96-101; Fig 1A)

Please explain how fever was defined? (What temperature cutoff?)

 Reply: Thanks for your comments. The definition of fever was added in the methods section. (Page 6, line 101-102)

Please explain what the follow-up period was. Did everyone get follow-up until they were either discharged from the hospital or died?

 Reply: Thanks for your comments, which we think is very valuable for the study. All patients enrolled in the study were followed-up until they were discharged or died, thus the follow-up period was the hospital days of everyone. The last patients in this cohort was discharged on March 3. (Page 5, line 82-84)

Results.

Please explain whether the 199 patients included all of the patients who were hospitalized at these five hospitals or whether this was a convenience sample.

 Reply: Thanks for your serious comments. We feel sorry that we did not provide enough information about whether the cohort was a convenience sample or not. The 199 patients included all the patients hospitalized at these five hospitals from Jan 22 to Feb 27, so this cohort was not a convenience sample. (Page 8, line 124-125). As there are eight SARS-CoV-2 designed hospitals in these five cities, not all patients with COVID-19 in above cities were included.

Line 124. Please provide a breakdown of BMI by normal weight, overweight, and obese

 Reply: Thanks for your kind suggestion. We’ve provide the detailed number of patients stratified by BMI according to the WHO recommendations for Body Mass Index (Page 8-9, Table 1)

Table 1.

Mild/common/severe/critical: Do the authors mean moderate rather than common?

 Reply: Thanks for your correction. We have revised it in the table, presenting the distribution of four clinical types of COVID-19. (Page 10, Table 1).

Did all 199 patients get a CT? Please add this information.

 Reply: Thanks for your comments. We have added the information that all 199 patients got a chest CT. (Page 10, line 143).

Line 138 and line 144. Please add a percentage to all numbers.

 Reply: Thanks for your reminding. The percentage of all numbers in both lines was added. (Page10, line 150 and line 156-157).

Line 151. Please phrase this association more clearly – was older age associated with lower lymphocyte counts?

 Reply: Thanks for your serious comments. We have clearly revised this description about the correlation between age and lymphocyte counts in our manuscript. (Page 12, line 162-165)

Line 161. Chinese medicine like Lianhuaqingwen accounted for 100%, 52.8%, 48.2%, and 40.2%, respectively. This sentence is not clear. Please rephrase.

 Reply: Thanks for your careful checks. We are sorry for our carelessness. We have revised the whole sentence as “patients receiving antiviral treatment, interferon inhalation, empirical antibiotics, and Chinese medicine, accounted for 100%, 52.8%, 48.2%, and 40.2%, respectively (Table 3)”. (Page 13, line 172-174)

Line 167. Aggravation – this term should be “deterioration in their clinical condition.” Aggravation does not seem like the appropriate word in English.

 Reply: Thanks for your correction. Based on your comments, we have corrected the “aggravation” as “deterioration” in the text. (Page 13, Table3; page 14, line 181)

Did any patients die? Please add this information to the results section.

Reply: Thanks for your comments. This information has been added in the section of results that only an 84-year-old man was died. (Page 14, line 185-186)

Table 4. Please describe the comparison for lymphocyte count, bilirubin and creatinine.

 Reply: Thanks for your comments. We’ve described the comparison for lymphocyte count, bilirubin and creatinine on their relationship with clinical severity of COVID-19 in our study (Page 14, line 193-195).

Discussion

Line 183

Please provide a specific comparison with numbers and percentages to support the claim that there were less severe patients and less deaths in the Fujian cohort compared to Wuhan.

Also, I am not convinced that a cohort of 199 patients is large enough to draw comparisons to the Wuhan about disease severity.

 Reply: Your suggestion really means a lot to us. According to your comments, we added a specific comparison with numbers and percentages to make our conclusion convincing (Page 15, line 200-203). In terms of mortality, only one death were reported in Fujian, far less than that of Wuhan where 48,137 cases were confirmed with 2132 (4.42%) deaths. (Page 15, line 204-206). Although the sample size of our cohort was not large enough to make comparison with cases in Wuhan, it enrolled more than 60% of the 296 infected patients confirmed in Fujian during the period, representing the situation of COVID-19 in Fujian to some extent.

In addition, the authors’ conclusions about early detection and quarantine measures being responsible for less severity are not supported by this article. The authors do not discuss incidence in this paper, and the authors also do not at all discuss quarantine measures that were implemented in Fujian province and how those compare to Wuhan. Finally, the cohort of 200 patients is quite small.

I suggest removing these broad conclusions and remaining with more modest conclusions.

 Reply: Thanks for your suggestions. We feel sorry for our overstatement of the conclusion. We agree with your view that the article could not reflect the relationship between disease severity and early detection or quarantine measures, so we have removed the related sentence in this Discussion section.

Line 192.

What is the connection of intestinal spread to family clusters? This is not logical. I suggest refocusing this paragraph on transmission and perhaps asymptomatic and presymptomatic transmission. The authors could cite other articles that described that most transmission occur in families.

 Reply: Thanks for your nice comments, which could make our manuscript more rigorous. We removed the description of intestinal transmission and refocused the asymptomatic and presymptomatic transmission of SRARS-CoV-2 by updating references, which have been revised in the manuscript. (Page 15, line 208-212)

Line 197.

The English needs to be reworked here

 Reply: Thanks for your suggestions. The sentence has been corrected as “Early studies of Wuhan showed that female patients accounted for 32% - 45.7% [14, 15, 21, 27], but no obvious gender proportion difference was found in our study”. (Page 16, line 216-217)

Line 214.

The authors should present data about Hepatitis B prevalence among cases in the results section if they intend to reference these data in the discussion section.

 Reply: Thanks for your kind reminding. We have added the description of the prevalence of Hepatitis B in the Results section. (Page 8, line 131-132)

Line 220.

Please update this paragraph to reflect the more recent findings that dexamethasone reduced mortality in severe COVID-19 patients and the changes in recommendations.

 Reply: Thanks for your suggestions, which is so helpful for the study. We provided the latest WHO recommendations on corticosteroid therapy for COVID-19 (Page 17, line 240-244), and added the latest findings about the effects of corticosteroid in COVID-19 by searching articles about the topic published recently in the Discussion section as follows: Another observational study enrolled 31 patients with COVID-19 found no association between corticosteroid use and virus shedding time, hospital length of stay, or duration of symptoms [35]. Moreover, corticosteroids treatment might be associated with increase mortality of critically ill patients [36]. The Randomized Evaluation of COVID-19 Therapy (RECOVERY) trial found that dexamethasone (6 mg/d, 10 days) reduced the 28-day all-cause mortality, especially most beneficial in patients who had symptoms for more than 7 days or required mechanical ventilation [37, 38]. In addition, considering high dose corticosteroid use closely associated with adverse events may be a risk factor of death [39], benefits and risks must be carefully weighed before commencing corticosteroid therapy, and the dosage and duration should be evaluated prudently. (Page 17, line 246-254).

There has been evidence of effectiveness of an antiviral drug, remdesivir, in reducing time to hospital discharge. The authors should update this paragraph to reflect these findings.

 Reply: Thanks for your comments. We have added the preliminary results of a clinical trial on remdesivir (Page 18, line 257-259). We also revised briefly the description about current development of vaccines for SARS-CoV-2. (Page 18, line 261-263)

Line 234.

The authors should not introduce new information in the discussion section. Information about course of virus positivity should be introduced in the results section and commented on in the discussion section.

 Reply: Thanks for your serious comments. The information about the one death in the study has been introduced in Results section and the nucleic acid results had been added in Results section, so we removed the related sentence in Page 17, line 234-238 of the original text. (Page 14, line185-186；Page 12, line 166),

Limitations

The authors should speak about how representative these hospitals are of hospitals in Fujian province. How many hospitals are there in Fujian? How many hospitalized cases of COVID-19 were there during the same period in other hospitals in Fujian?

 Reply: Thanks for your serious comments. We added information about the representativeness of our cohort in the manuscript as follows: 296 indigenous cases were confirmed and hospitalized in nineteen designed hospitals for SARS-CoV-2 in Fujian province during the epidemic. Herein only 199 cases with COVID-19 in five hospitals were enrolled, so our conclusions maybe not represent the true situation of the outbreak in entire Fujian Province. (Page 18, line 265-269)

Conclusions paragraph

The authors should be cautious about drawing conclusions relative to Wuhan because of the small sample size.

Reply: We sincerely appreciate the valuable comments. Although the sample size is not large enough to be compared with that of Wuhan, 199 of 296 patients (67.2%) were enrolled in the study during the same period in Fujian province, so we could make prudent conclusions that compared with the COVID-19 in Wuhan during the early outbreak, most COVID-19 patients in Fujian Province were non-severe cases and had a lower mortality rate to some extent. (Page 18, line 271-273)

We have refer to Table 3 in the text. (Page 13, line 176; page 14, line 183)

In addition, the funding information has been removed from the manuscript.

Here we must express our gratitude to you and the reviewers once again, for giving us this chance to revise. All of these comments have contributed a lot to improve the quality of our article. After this revision, we have written a point-by-point response letter to you as you can see above. We feel sorry for our poor writings and tried our best to improve the manuscript and made some changes in the manuscript. These changes will not influence the content and framework of the paper. We hope the correction will meet with approval. We really appreciate for your warm work earnestly, and would be glad to respond to any further questions and comments that you may have.

Thank you once again!.

---

## [Editor Report · Decision Letter 1]

14 Sep 2020

PONE-D-20-18174R1

Clinical characteristics of 199 discharged patients with COVID-19 in Fujian province: a multicenter retrospective study between January 22nd and February 27th, 2020

PLOS ONE

Dear Dr. shao,

Thank you for submitting your manuscript to PLOS ONE. After careful consideration, we feel that it has merit but does not fully meet PLOS ONE’s publication criteria as it currently stands. Therefore, we invite you to submit a revised version of the manuscript that addresses the points raised during the review process.

The authors do not appear to have addressed any of Reviewer 1's comments. The authors should revise the manuscript and include point-by-point responses to Reviewer 1's comments, which they can add to the "response to reviewers" document. 

We look forward to receiving your revised manuscript.

Kind regards,

Mark Katz

Academic Editor

PLOS ONE

---

## [Author Response · Author response to Decision Letter 1]

2 Oct 2020

Dear Editors and Reviewers,

Thank you for your letter and for the reviewers’ comments concerning our manuscript entitled “Clinical characteristics of 199 discharged patients with COVID-19 in Fujian province: a multicenter retrospective study” (ID: PONE-D-20-18174). Those comments are all valuable and very helpful for revising and improving our paper, as well as the important guiding significance to our researches. We have studied comments carefully and have made correction which we hope would meet with approval. Revised portion are marked in yellow in the paper. The main corrections in the paper and responds to reviewer’s comments are as following:

Abstract:

Intro: The authors should add some background information on the COVID-19 outbreak in China and when the outbreak started in Fujian, and what is known until now about the outbreak in Fujian.

Methods: The authors should describe what was done in the methods but the number of patients should be first introduced in the results section. In the methods section please include the dates of the study and explain during what part of the outbreak the data were collected. Were these the first hospitalized cases in the province or was this a convenience sample from the middle of the outbreak?

Results: Please define severe and critical in the methods section. Line 35 please describe more clearly the relationship between age and lymphopenia and oxygen. The older the person the lower the lymphocyte count and the lower the oxygen level? Please describe this in the text.

Conclusions: Mention the findings about severity in Wuhan so that the reader can understand what was different here.

 Reply: Thank you for your kind comments. We have added the background information of COVID-19 outbreak in China and Fujian province (Page 2, line 24-26). The dates of the study was added and the number of patients was deleted in Methods section (Page 2, line 30-31), and all cases were the first hospitalized cases in these cities (Page 2, line 29). The definition of four types of COVID-19 was introduced in Methods section in manuscript (Page 6, line 96-102), and the description of relationship between age and lymphopenia and oxygen was rephrased clearly (Page 2, line 39-40). The early COVID-19 mortality in Wuhan was mentioned in the Conclusions section (Page 3, line 43).

Manuscript:

Title: Add dates of the study

 Reply: Thanks for your serious comments. We have added dates of the study into the title. (Page 1, line 2-3)

Intro:

Please briefly the history of the COVID-19 epidemic of Fujian province and also the characteristics of the provine. When was the first case identified. How many cases have been identified until now? Where is Fujian province in China in relation to Wuhan and what is the population?

 Reply: Thanks for your comments. According to data published by the Centers for Disease Control of Fujian province, we introduced briefly when and where the first COVID-19 case was confirmed in Fujian, and the total number of indigenous cases (Page 5, line 68-71). We also described the geographical location and population of Fujian province (Page 4, line 67-68)

Methods:

Line 78. The ethics statement should be placed at the end of the methods section.

 Reply: Thanks for your suggestion. We have placed the ethics statement at the end of methods section behind the Statistical Analysis. (Page 7, line 119-122)

Definitions.

Please provide the definition for moderate patients.

 Reply: Thanks for your reminding. We apologize for our negligence regarding confusing the moderate type and common type. We have corrected “common type” as “moderate type” and reintroduce the definition of four types of COVID-19 patients respectively. In addition, we also revised that in Fig 1A (Page 6, line 96-101; Fig 1A)

Please explain how fever was defined? (What temperature cutoff?)

 Reply: Thanks for your comments. The definition of fever was added in the methods section. (Page 6, line 101-102)

Please explain what the follow-up period was. Did everyone get follow-up until they were either discharged from the hospital or died?

 Reply: Thanks for your comments, which we think is very valuable for the study. All patients confirmed were followed-up until they were either discharged or died, thus the follow-up period was exactly the hospital days of everyone. The last patients in this cohort was discharged on March 3. (Page 5, line 82-84)

Results.

Please explain whether the 199 patients included all of the patients who were hospitalized at these five hospitals or whether this was a convenience sample.

 Reply: Thanks for your serious comments. We feel sorry that we did not provide enough information about whether the cohort was a convenience sample or not. The 199 patients included all the patients hospitalized at these five hospitals from Jan 22 to Feb 27, so this cohort was not a convenience sample. (Page 8, line 125-126). As there are eight SARS-CoV-2 designed hospitals in these five cities, not all patients with COVID-19 in above cities were included.

Line 124. Please provide a breakdown of BMI by normal weight, overweight, and obese

 Reply: Thanks for your kind suggestion. We’ve provide the detailed number of patients stratified by BMI according to the WHO recommendations for Body Mass Index (Page 8-9, Table 1)

Table 1.

Mild/common/severe/critical: Do the authors mean moderate rather than common?

 Reply: Thanks for your correction. We have revised it in the table, presenting the distribution of four clinical types of COVID-19. (Page 10, Table 1).

Did all 199 patients get a CT? Please add this information.

 Reply: Thanks for your comments. We have added the information that all 199 patients got a chest CT. (Page 10, line 144).

Line 138 and line 144. Please add a percentage to all numbers.

 Reply: Thanks for your reminding. The percentage of all numbers in both lines was added. (Page 10, line 151; page 12 line 157-158).

Line 151. Please phrase this association more clearly – was older age associated with lower lymphocyte counts?

 Reply: Thanks for your serious comments. We have clearly revised this description about the correlation between age and lymphocyte counts in our manuscript. (Page 12, line 164-165)

Line 161. Chinese medicine like Lianhuaqingwen accounted for 100%, 52.8%, 48.2%, and 40.2%, respectively. This sentence is not clear. Please rephrase.

 Reply: Thanks for your careful checks. We are sorry for our carelessness. We have revised the whole sentence as “patients receiving antiviral treatment, interferon inhalation, empirical antibiotics, and Chinese medicine, accounted for 100%, 52.8%, 48.2%, and 40.2%, respectively (Table 3)”. (Page 13, line 172-174)

Line 167. Aggravation – this term should be “deterioration in their clinical condition.” Aggravation does not seem like the appropriate word in English.

 Reply: Thanks for your correction. Based on your comments, we have corrected the “aggravation” as “deterioration” in the text. (Page 13, Table3; page 14, line 182; page 7, line 106-107)

Did any patients die? Please add this information to the results section.

Reply: Thanks for your comments. This information has been added in the section of results that only an 84-year-old man was died. (Page 14, line 186-187)

Table 4. Please describe the comparison for lymphocyte count, bilirubin and creatinine.

 Reply: Thanks for your comments. We’ve described the comparison for lymphocyte count, bilirubin and creatinine on their relationship with clinical severity of COVID-19 in our study (Page 15, line 195-196).

Discussion

Line 183

Please provide a specific comparison with numbers and percentages to support the claim that there were less severe patients and less deaths in the Fujian cohort compared to Wuhan.

Also, I am not convinced that a cohort of 199 patients is large enough to draw comparisons to the Wuhan about disease severity.

 Reply: Your suggestion really means a lot to us. According to your comments, we added a specific comparison with numbers and percentages to make our conclusion convincing (Page 15, line 201-204). In terms of mortality, only one death were reported in Fujian, far less than that of Wuhan where 48,137 cases were confirmed and 2132 (4.42%) deaths. (Page 15, line 204-206). 

Although the sample size of our cohort was not large enough to make comparison with cases in Wuhan, it enrolled 199 cases (＞60%) of the 296 infected patients confirmed in Fujian during the period, representing the situation of COVID-19 in Fujian to some extent.

In addition, the authors’ conclusions about early detection and quarantine measures being responsible for less severity are not supported by this article. The authors do not discuss incidence in this paper, and the authors also do not at all discuss quarantine measures that were implemented in Fujian province and how those compare to Wuhan. Finally, the cohort of 200 patients is quite small.

I suggest removing these broad conclusions and remaining with more modest conclusions.

 Reply: Thanks for your suggestions. We feel sorry for our overstatement of the conclusion. We agree with your view that the article could not reflect the relationship between disease severity and early detection or quarantine measures, so we have removed the related sentence in this Discussion section.

Line 192.

What is the connection of intestinal spread to family clusters? This is not logical. I suggest refocusing this paragraph on transmission and perhaps asymptomatic and presymptomatic transmission. The authors could cite other articles that described that most transmission occur in families.

 Reply: Thanks for your nice comments, which could make our manuscript more rigorous. We removed the description of intestinal transmission and refocused the asymptomatic and presymptomatic transmission of SRARS-CoV-2 by updating references, which have been revised in the manuscript. (Page 15, line 208-213)

Line 197.

The English needs to be reworked here

 Reply: Thanks for your suggestions. The sentence has been corrected as “Early studies of Wuhan showed that female patients accounted for 32% - 45.7% [14, 15, 21, 27], but no obvious gender proportion difference was found in our study”. (Page 16, line 217-218)

Line 214.

The authors should present data about Hepatitis B prevalence among cases in the results section if they intend to reference these data in the discussion section.

 Reply: Thanks for your kind reminding. We have added the description of the prevalence of Hepatitis B in the Results section. (Page 8, line 132-133)

Line 220.

Please update this paragraph to reflect the more recent findings that dexamethasone reduced mortality in severe COVID-19 patients and the changes in recommendations.

 Reply: Thanks for your suggestions, which is so helpful for the study. We provided the latest WHO recommendations on corticosteroid therapy for COVID-19 (Page 17, line 241-245), and added the latest findings about the effects of corticosteroid in COVID-19 by searching articles about the topic published recently. (Page 17, line 247-255).

There has been evidence of effectiveness of an antiviral drug, remdesivir, in reducing time to hospital discharge. The authors should update this paragraph to reflect these findings.

 Reply: Thanks for your comments. We have added the preliminary results of a clinical trial on remdesivir (Page 18, line 258-260). We also revised the description about current development of vaccines for SARS-CoV-2. (Page 18, line 2612-264)

Line 234.

The authors should not introduce new information in the discussion section. Information about course of virus positivity should be introduced in the results section and commented on in the discussion section.

 Reply: Thanks for your serious comments. The information about the one death in the study has been introduced in Results section and the nucleic acid results had been added in Results section, so we removed the related sentence in Page 17, line 234-238 of the original text. (Page 14, line185-186; page 12, line 166),

Limitations

The authors should speak about how representative these hospitals are of hospitals in Fujian province. How many hospitals are there in Fujian? How many hospitalized cases of COVID-19 were there during the same period in other hospitals in Fujian?

 Reply: Thanks for your serious comments. We added information about the representativeness of our cohort in the manuscript as follows: 296 indigenous cases were confirmed and hospitalized in nineteen designed hospitals for SARS-CoV-2 in Fujian province during the epidemic. Herein only 199 cases with COVID-19 in five hospitals were enrolled, so our conclusions maybe not represent the true situation of the outbreak in entire Fujian Province. (Page 18, line 266-272)

Conclusions paragraph

The authors should be cautious about drawing conclusions relative to Wuhan because of the small sample size.

Reply: We sincerely appreciate the valuable comments. Although the sample size is not large enough to be compared with that of Wuhan, 199 of 296 patients (67.2%) were enrolled in the study during the same period in Fujian province, so we could make prudent conclusions that compared with the COVID-19 in Wuhan during the early outbreak, most COVID-19 patients in Fujian Province were non-severe cases and had a lower mortality rate to some extent. (Page 18, line 273-275)

We have refer to Table 3 in the text. (Page 13, line 173; page 14, line 181)

In addition, the funding information has been removed from the manuscript.

Response to Review 1’ comments

Reviewer #1: This manuscript is a descriptive retrospective analysis of 199 COVID-19 patients admitted to one of five tertiary care centers in Fujian province, China between January 22 and February 27, 2020. A secondary aim of the study was to identify factors associated with disease severity. Study inclusion criteria and statistical analyses are appropriate. The data collection represents a relatively short period of observation, which may influence some of the study findings. Authors state that all data are available in the manuscript and additional files.

Reply: Thanks for your serious comments. Since we only collected data of these patients during hospitalization, the period of observation was dependent of the length of hospital stay (Page 5, line 82-84). In this study, we mainly described the early outbreak condition of COVID-19 in Fujian province, China, so the period of observation may not influence the findings in study to a large degree. 

Methods:

Additional description of how severity categories were collapsed for multivariable logistic regression is warranted in the methods section. It appears that severe/critical cases were combined and compared to mild/moderate severity illnesses. But this should be explicitly stated.

 Reply: Thanks for your serious comments, which was very valuable for our study. We have added the severity categories in the Statistical Analysis. According to the highest type during hospitalization, patients were divided into two groups: non-severe (mild and moderated type) cases and severe (severe and critical type) cases. (Page 7, line 112-114; page 14, line 180-181). 

Results:

In the description of treatment (text and table) it might be of interest to report which antivirals were used for treatment I see this is mentioned in the discussion, but could be noted in the results as well.

Reply: Thanks for your suggestions. For all patients, we recounted and presented the main antiviral drugs used in the cohort in the text: The antiviral drugs were mainly lopinavir/ritonavir, arbidol, ribavirin and oseltamivir. (Page7, line 174) and their exact usage percentage in Table 3. 

Discussion:

On page 17 line 244-245 the authors state "compared with patients infected with COVID-19 in Wuhan, our study suggested most patients in Fujian province were mild and moderate with a minority of severe cases". I think this overstates the results. In this relatively small study of hospitalized patients for a short period (4 weeks) there were 33/199 (16.6%) that were severe, which is in line with estimates from other locations. Case fatality was lower for the observation period in this study compared to others. I'm not convinced the follow-up was long enough to accurately determine the true extent of severe illness and mortality (as data collection ended on March 3, 2020). Readmission and deterioration have been issues in other places. Other the conclusions follow from the results. Some discussion of the implications of the short follow-up time is warranted.

Reply: Your suggestion really means a lot to us. The conclusion on comparison between early outbreak in Wuhan and Fujian province was indeed hasty and a bit overstated, so we referred to detailed numbers and percentages in published articles and official data to make the conclusion convincing (Page 15, line 201-206). In this study, we tracked closely the dynamic change of each patient during hospitalization until they were died or cured and discharged, so the true extent of severe illness could be assessed accurately (Page 5, line 82-84). Although the follow-up time limited to length of hospital stay was only five weeks (Jan 22- March 3), it didn’t influence our main findings in the study. Of course, changes in these cured patients’ condition after discharge, such as relapse and deterioration, need longer follow-up to assess. As for the mortality, except one death, all the other patients infected with COVID-19 were cured and met the discharge standard before March 3 (Page 7, line 107-109), so the follow-up was enough to help us obtain the real mortality of COVID-19 in these hospitals. Considering that the readmission and deterioration have been reported in other places, we will next follow up these discharged patients to determine their readmission, deterioration and even sequela. Discussion of the implications of the short follow-up time is implicated in the manuscript. (Page 18, line 270-272). 

Here we must express our sincere gratitude to you and the reviewers once again, for giving us chances to revise. All of these comments have contributed a lot to improve the quality of our article. After this revision, we have written a point-by-point response letter to you as you can see above. We feel sorry for our poor writings and tried our best to improve the manuscript and made some changes in the manuscript. These changes will not influence the content and framework of the paper. We hope the correction will meet with approval. We really appreciate for your warm work earnestly, and would be glad to respond to any further questions and comments that you may have.

Thank you once again!.

---

## [Editor Report · Decision Letter 2]

8 Oct 2020

PONE-D-20-18174R2

Clinical characteristics of 199 discharged patients with COVID-19 in Fujian province: a multicenter retrospective study between January 22nd and February 27th, 2020

PLOS ONE

Dear Dr. shao,

Thank you for submitting your manuscript to PLOS ONE. After careful consideration, we feel that it has merit but does not fully meet PLOS ONE’s publication criteria as it currently stands. Therefore, we invite you to submit a revised version of the manuscript that addresses the points raised during the review process.

The authors have adequately addressed nearly all of the comments from the reviewers. 

The manuscript is still suffering from many writing errors. If the authors could have a native English speaker and writer to review the manuscript for flow, subject-noun agreement, and to ensure that everything is written in the past tense, the manuscript would then be more suitable for publication.

In addition, the authors should make the following changes:

Because you did not do a statistical comparison, please change line 203 to “suggesting that the early outbreak in Fuian may have been milder than that in Wuhan.” (instead of the current wording "indicating..")

Please update the section on corticosteroids in the discusson section to include recently published data in JAMA supporting the use of corticosteroids in severe and critical COVID-19 patients and removing less relevant publications on corticosteroids.

We look forward to receiving your revised manuscript.

Kind regards,

Mark Katz

Academic Editor

PLOS ONE

---

## [Author Response · Author response to Decision Letter 2]

20 Oct 2020

The manuscript is still suffering from many writing errors. If the authors could have a native English speaker and writer to review the manuscript for flow, subject-noun agreement, and to ensure that everything is written in the past tense, the manuscript would then be more suitable for publication.

Reply: Your suggestions really means a lot to us. We feel sorry for our poor writings, and we have invited a native English speaker and writer from USA to help correct and polish our article carefully.

In addition, the authors should make the following changes:

Because you did not do a statistical comparison, please change line 203 to “suggesting that the early outbreak in Fujian may have been milder than that in Wuhan.” (instead of the current wording "indicating..")

Reply: Thanks for your kind suggestion. We have changed the sentence in line 203 to “suggesting that the early outbreak in Fujian may have been milder than that in Wuhan.” (Page 16, line 206-207).

Please update the section on corticosteroids in the discusson section to include recently published data in JAMA supporting the use of corticosteroids in severe and critical COVID-19 patients and removing less relevant publications on corticosteroids.

Reply: Thanks for your serious suggestion. We have added the newly results published in JAMA on the administration of corticosteroids in critical ill patients with COVID-19 (Page 18, line 252-254) and less relevant publications was removed in the manuscript.

---

## [Editor Report · Decision Letter 3]

2 Nov 2020

Clinical characteristics of 199 discharged patients with COVID-19 in Fujian province: a multicenter retrospective study between January 22nd and February 27th, 2020

PONE-D-20-18174R3

Dear Dr. shao,

We’re pleased to inform you that your manuscript has been judged scientifically suitable for publication and will be formally accepted for publication once it meets all outstanding technical requirements.

Kind regards,

Mark Katz

Academic Editor

PLOS ONE
---

## [Editor Report · Acceptance letter]

5 Nov 2020

PONE-D-20-18174R3 

Clinical characteristics of 199 discharged patients with COVID-19 in Fujian Province: a multicenter retrospective study between January 22nd and February 27th, 2020 

Dear Dr. shao:

I'm pleased to inform you that your manuscript has been deemed suitable for publication in PLOS ONE. Congratulations! Your manuscript is now with our production department. 

Kind regards, 

on behalf of

Dr. Mark Katz 

Academic Editor

PLOS ONE